# A Dense Neural Network Approach for Detecting Clone ID Attacks on the RPL Protocol of the IoT

**DOI:** 10.3390/s21093173

**Published:** 2021-05-03

**Authors:** Carlos D. Morales-Molina, Aldo Hernandez-Suarez, Gabriel Sanchez-Perez, Linda K. Toscano-Medina, Hector Perez-Meana, Jesus Olivares-Mercado, Jose Portillo-Portillo, Victor Sanchez, Luis Javier Garcia-Villalba

**Affiliations:** 1Instituto Politecnico Nacional, ESIME Culhuacan, Mexico City 04440, Mexico; cmoralesm1702@alumno.ipn.mx (C.D.M.-M.); gasanchezp@ipn.mx (G.S.-P.); ltoscano@ipn.mx (L.K.T.-M.); hmperezm@ipn.mx (H.P.-M.); jolivares@ipn.mx (J.O.-M.); jportillop@ipn.mx (J.P.-P.); 2Department of Computer Science, University of Warwick, Coventry CV4 7AL, UK; v.f.sanchez-silva@warwick.ac.uk; 3Group of Analysis, Security and Systems (GASS), Department of Software Engineering and Artificial Intelligence (DISIA), Faculty of Computer Science and Engineering, Office 431, Universidad Complutense de Madrid (UCM), Calle Profesor José García Santesmases, 9, Ciudad Universitaria, 28040 Madrid, Spain; javiergv@fdi.ucm.es

**Keywords:** Clone ID attack, deep learning, Internet of Things, IoT, intrusion detection, IDS, machine learning, RPL

## Abstract

At present, new data sharing technologies, such as those used in the Internet of Things (IoT) paradigm, are being extensively adopted. For this reason, intelligent security controls have become imperative. According to good practices and security information standards, particularly those regarding security in depth, several defensive layers are required to protect information assets. Within the context of IoT cyber-attacks, it is fundamental to continuously adapt new detection mechanisms for growing IoT threats, specifically for those becoming more sophisticated within mesh networks, such as identity theft and cloning. Therefore, current applications, such as Intrusion Detection Systems (IDS), Intrusion Prevention Systems (IPS), and Security Information and Event Management Systems (SIEM), are becoming inadequate for accurately handling novel security incidents, due to their signature-based detection procedures using the matching and flagging of anomalous patterns. This project focuses on a seldom-investigated identity attack—the Clone ID attack—directed at the Routing Protocol for Low Power and Lossy Networks (RPL), the underlying technology for most IoT devices. Hence, a robust Artificial Intelligence-based protection framework is proposed, in order to tackle major identity impersonation attacks, which classical applications are prone to misidentifying. On this basis, unsupervised pre-training techniques are employed to select key characteristics from RPL network samples. Then, a Dense Neural Network (DNN) is trained to maximize deep feature engineering, with the aim of improving classification results to protect against malicious counterfeiting attempts.

## 1. Introduction

The Internet of Things (IoT) is a network comprised of smart devices, equipped with embedded sensors, actuators, processors, and transceivers [1], mostly connected to an application or server over the Internet. A recent study [2] mentioned that, by 2020, IoT technology will be available in 95% of the electronic circuitry of newly emerging products, comprising approximately twenty billion smart devices in constant use. Moreover, a report presented by Cisco [3] demonstrated that, due to the rapid evolution and popularity of IoT gadgets and wearables, the amount of fully connected devices around the world may be as high as fifty billion. Due to the interoperability of the IoT paradigm, Wireless Sensor Networks (WSN) have been sketched as a subset of IoT technology, in which each device can employ inherent sensors to work as a mesh network.

At present, the applications of IoT and WSNs are predominantly being engaged for industrial needs; as an example, in [4], WSNs were used to monitor power transmission lines, in order to reduce the damage caused by natural disasters and extreme weather conditions, by collecting data from different types of sensors distributed on transmission towers. On the other hand, in [5], cluster-based data aggregation methods were executed in different-sized WSNs, in order to gauge configurations and identify pests in coffee plantations. Furthermore, in [6], a set of WSNs was arranged to measure earth vibrations and to monitor the structural health of constructions. One of the most important (and often critical) form of industrialized WSNs is in aircraft manufacturing: Figure 1 illustrates Aircraft Strength Testing (AST), an auditing procedure that exploits linked sensors to assess structural damage, such as fatigue cracks, part rigidity, corrosion, and so on [7].

By their very nature, IoT and WSNs are likely to crash due to packet transmission loss and low network throughput—constraints that are being progressively amended by standardizing different routing protocols [8], among which the Routing Protocol for Low-Power and Lossy Networks protocol (RPL) stands out, due to its rapid flexibility and efficient routing—characteristics required to deal with different network technologies, smart IP addresses, the enabling of IoT nodes, and quality of service (QoS) support [8,9]. In view of its noticeable advantages, RPL is becoming the underlying technology for many IoT device manufacturers and vendors in a wide range of application domains; however, it also provides an entry point for security threats that compromise legacy structures that are implicit in RPL networks, such as the Low Power and Lossy Networks (LLNs). Data collection, sharing, transparency, updating, and secure communication enforcing are not just the only inquiries for IoT and WSNs protocols to reach security goals, but also how protection tactics and strategies are being modeled to offer reliable end-to-end solutions. A lack of security controls, mechanisms, and incident monitoring capabilities can expose distinct weaknesses across the network layers of RPL and IoT architectures. Firstly, at the network layer, common vulnerabilities include eavesdropping, packet sniffing, routing attacks, and Denial of Service (DoS); the latter is possible thanks to the high accessibility to smart devices running with default configuration, with hardly any security measures implemented [10]. At the application layer, frequent attacks include phishing, malicious code and object injection, and Cross-Site-Request-Forgery (CSRF), as well as any risk concerning web-based applications.

As explained in [11], RPL and LLN security is an extensive subject of study; however, certain breaches remain unaddressed: Routing falsification, Clone ID, and Byzantine intrusion. In addition, in [12], it was mentioned that perimeter defensive systems must update their detection controls towards major routing attacks, such as sinkhole, blackhole, selective forwarding, sybil and, specifically, identity cloning. A summary of cyber-attack types directed at IoT protocols, emphasizing those aimed at RPL and LLNs, is depicted in Figure 2.

Detection/prevention mechanisms work through observations of network traffic, by means of IDS, IPS, firewall, and Security Information and Event Management (SIEM) systems. The aforementioned are physical or logic implementations working on a network, host, or hybrid environments and, for instance, are configured with rules, policies, and/or behavioral- and anomaly-based signatures [13]. Indeed, monitoring network flows for malicious signature matching, rules/policy disruption, and flagging abnormal patterns can lead to time and resource consumption, an increase in high false/positive rates, limited customization, and constant database updating, making this last factor a significant setback, as intruders are constantly crafting novel and more sophisticated artifacts [14].

Although the efforts to mitigate the previously mentioned flaws have been addressed by a comprehensive number of secure protocols [15], the proposed mechanisms are still not entirely suitable for IoT resource-constrained ecosystems with novel and unconventional attack surfaces, affecting the integrity of routing algorithms and the trustworthiness of RPL node identity. In that sense, Wallgren et al. [16] mentioned that, due to the misuse of the Directed Acyclic Graph (DAG)—the core engine for RPL topology organization—attackers can easily bypass voting schemes by counterfeiting the identities of legitimate IoT physical nodes. This type of identity impersonation, better known as the Clone ID attack, is challenging to overcome, considering that RPL algorithms are bound to multiple metrics to determine each node level and ranking within a network, opening the possibility of a worst-case scenario: when the root node, as it is known by the Destination-Oriented Directed Acyclic Graph (DODAG) is compromised. As a result, any data sent through a cloned root node will be directly available to an attacker and might be susceptible to alteration, leaking, spoofing, and ex-filtration, while the cloned node and siblings can remain unreachable, levering transmission inconsistencies over the entire network. ID Cloning attacks have been cited in early work [17,18]; however, the remediation and mitigation methods have only been partially disclosed and have been limited to node ranking optimization, secure node election mechanisms, and malicious node elimination through blacklisting additions to IDS/IPS rules.

Consequently, more intelligent and adaptable schemes are being developed to expand the detection capabilities of classical security defense systems. In that sense, the incorporation of Artificial Intelligence (AI), Machine Learning (ML), Deep Learning (DL), and Reinforcement Learning (RL) sensors has become imperative to deal with large amounts of samples, maximize feature engineering, learn from latent abnormal patterns, reduce the time for disclosing unknown vulnerabilities, and reinforce classification outcomes [19]. Despite this, AI-enhanced and semi-enhanced frameworks are more oriented towards easing flooding attacks [19,20], wormhole attacks [21], machine-driven hello-flood attacks [22], Received Signal Strength Indicator (RSSI) flooding attacks [23], and DDoS attacks [24], leaving Cloning ID attacks as a virtually unexplored area of research. In this work, we present a novel protection framework based on unsupervised ML pre-processing techniques, along with a Dense Neural Network (DNN) approach, to effectively detect counterfeiting attacks on RPL-based network conversations.

### Aiming and Research Contributions

The RPL protocol has been the subject of several studies since its creation and implementation. Like any type of standard, IoT device manufacturers and application developers have adapted new technologies based on the premises of RPL, in which information security has not been a primary point in its design. Over time, many vulnerabilities have been reported, which have been fully or partially remediated, according to their level of risk. One of the most difficult threats to counter is the Clone ID Attack, as, according to [25], this type of threat is independent of the applications supported in WSNs, where stealthily malicious actors are able to impersonate multiple identities in the absence of default authorization mechanisms. With the rise of the information security landscape, corrective controls have been proven to be not entirely effective, as centralized and distributed methods are needed, including the use of cryptographic algorithms, hardware modifications, high-cost resources to evaluate sensor nodes positions, neighbourhood analysis and, above all, the ability to operate in restricted environments, such as IoT. On the other hand, the addition of physical or logical perimeter systems must allow for the in-depth filtering of the content of RPL messages, where the rules and detection policies may not be adequate, leaving a large gap in the writing of a signature and the craft of a new threat. This means that there are no intelligent or adaptive controls for most of these applications [26]. As mentioned in Section 1, one of the novel approaches that can detect a Clone ID pattern is the incorporation of AI techniques, which can better characterize the behaviour of an attack and present a faster detection, supported by performance measures that endorse the functionality of the algorithms used. The contribution of this paper highlights this importance, focusing on a robust solution and implementation as a possible framework for an IDS/IPS appliance. Although there have been works that presented AI as an alternative choice (specifically, ML or DL) to solve several vulnerabilities in RPL, their scope was limited to threats in areas such as routing, localization, data aggregation, and synchronization. Moreover, the proposals had procedures limited to shallow algorithms, with little capacity for the extraction and selection of important features. To date, it is unknown if there is any other work using DL to tackle the Clone ID Attack. The contributions of this article are listed below:One of the first detection frameworks for Clone ID Attack based on Artificial Intelligence algorithms is developed; the construction is based on the following premises:The pre-processing of traffic samples obtained by simulations with real traffic from IoT and WSN sensors, which can filter, scale, and reduce the complexity of the samples;The use of low-cost feature selection and extraction techniques, in order to ideally represent key evidence resulting from an attack and regular behaviours over a WSN.The presentation of a possible real deployment detection scenario, taking into account the on-premise capabilities and constraints of current IDS/IPS systems, as well as the implications of using an ML-based solution.

The rest of this paper is organized as follows: Section 2 provides a review of the RPL protocol and a synopsis of identity cloning attacks. Section 3 describes the related work. In Section 4, the intended framework is detailed, as well as data gathering, pre-processing, and DNN training tasks. Section 5 presents the experimental results and discussions. Finally, Section 6 concludes this work.

## 2. The RPL Protocol

The IoT is a compound of multiple linked and interoperable devices, majorly framed with limited resources and lossy radio-links. To grant a successful management and routing control, two main protocols are used in IoT network layers: IPv6 over Low-Power 6LoWPAN (WPAN) and RPL. Initially, the 6LoWPAN protocol was standardized by The Internet Engineering Task Force (IETF), in order to allow IPv6 packets to be carried efficiently in small link layer frames in WSNs backgrounds [27].

In its more recent adaptations, 6LoWPAN has been used over broadened networking media, such as Low-Power Radio Frequency (LPRF), Bluetooth Smart (BS), Power Line Control (PLC), and Low-Power Wi-Fi (LPWIFI). To drive forward different routing paths on Low-power and Lossy Networks (LLNs), such as point-to-point (P2P), point-to-multipoint (P2MP), and multipoint-to-point (MP2P), the RPL protocol has been established as a more resilient alternative, by creating a Destination-oriented Directed Acyclic Graph (DODAG) between the IoT nodes and 6LoWPAN unidirectional traffic. To assemble incoming network nodes, DODAG senses and ranks each node by indicating its relative position with respect to others and with respect to the DODAG itself. As a case in point, in Figure 3, the sensor nodes A and C are parents of nodes D, E, and F, while node F acts as the parent of node G.

As with any network protocol, RPL uses a control message medium to identify requested and forwarded packets. The message body is a DODAG object composed of ICMPv6 code fields, as well as source/destination addresses for all nodes in a hierarchical structure. In Table 1, the RPL control message types are listed [28].

From Table 1, it is worth noticing two objects that play a principal role in DODAG organization: The DIS control message, which is responsible for sensing, discovering, and requesting neighbouring nodes to join a DODAG tree; and the DIO control message, for assigning an identity to a newly added node and hierarchically reconstructing the DODAG.

### 2.1. Clone ID Attack on the RPL Protocol

In the process of sensing RPL-based connections, the DIS and DIO control messages are liable to be manipulated or created to commit identity counterfeiting—namely, the Clone ID attack—in which malicious actors learn about the configuration of DODAG trees and modify the information required to add a malicious node. This attack is possible because the RPL standard version is not intended to implement security and self-healing mechanisms by default. In fact, the IEFT strongly recommends data confidentiality and integrity configurations using pre-shared key authentication. These characteristics can increase the reliability of the resources, but its implementation has a severe impact, in terms of limiting the nodes mobility and network performance. It has also been demonstrated that an attacker inside the network can easily bypass security controls, due to the complexity of self-configuration and organization between nodes, making key management and peering a major complication. On this basis, nodes can join freely without enforcing any authorization or authentication policy, opening a great opportunity to replicate multiple nodes in different locations [29]. In Figure 4a,b, the steps to conduct a Clone ID attack on a RPL default environment are pictured: 1. A malicious actor senses a RPL-based network and targets it, then the DODAG configuration is read and a malicious node *Z* learns from sibling identities and impersonate a selected one, *E*. In this stage, *Z* remains a sibling node with no competition with the targeted node; 2. Forged DIS control messages containing the cloned identity of the sibling node, *E*, are sent by the malicious node, *Z*, until access is granted to the network; 3. Therefore, when the malicious node, *Z*, is accepted as a member of the graph, all the other nodes in the network send a DIO control message to reconstruct the DODAG topology and communication with the malicious node, *Z*, begins in a transparent way. It is important to mention that the legitimate node, *E*, may send another DIS message in order to restore communication with the network, resulting in a continuous competition between the malicious node, *Z*, and the legitimate node *E*, in order to gain trustworthiness among the graph members, in such a way that the winner will be the one who can maintain DIS messages for a longer span of time. As described by [30], this is possible due to two main factors: 1. The IEFT states that the RPL protocol uses by default a mode called *Unsecured*, where the basic messages DIS, DIO, DAO and DAO-ACK can be easily modified, without any integrity nor authenticity check. This is a great advantage for the attacker to impersonate a node within the network. 2. RPL allows any node to join the DODAG tree at any time, by means of DIS and DIO messages to incorporate itself and re-arrange the DODAG tree, as mentioned before. With this, an illegitimate node will take advantage of the Trickle Timer mechanism of the protocol, which periodically sends DIO messages to check the stability of the DODAG tree and thus, generate persistence within the DODAG tree.

Internally, RPL sensing algorithms are not aware of node identities, nor are security access control lists, mechanisms, and verification schedules mandatory to affiliate to a DODAG tree. Additionally, the geographical information of each node is not deemed to track its position, according to its node parent hierarchy, such that a single cloned node can be distributed in multiple leaves on the DODAG graph. Crafting and deploying a Clone ID Attack is directly linked to the absence of security methods contained in the implementation of RPL on conventional devices. With this in mind, sensors are susceptible to physical capture attacks, such that adversaries can easily program an unlimited number of replicated nodes by spoofing the WSN. In agreement with the taxonomies presented in [31], node replication is an application-independent attack over IoT networks that can be reproduced in static or mobile ways, depending on the WSN architecture. In a static manner, each node is randomly created with a fixed geographical position inside a DODAG graph; in counterpart, in the mobile form, nodes can move through dynamic routing. The steps to reproduce a Clone ID Attack, either static or mobile, are listed as follows: 1. A malicious actor inside a vulnerable WSN captures sensor nodes physically; 2. A selected legal node is isolated in a span of time from the network; 3. The node is analyzed and sensible information is collected (node id, secret keys, cryptographic data, and so on); 4. The attacker replicates the selected node by counterfeiting its identity and mounts it in key geographical positions. Additionally, an intruder can decide if is worth replicating multiple nodes in the network and place them in other positions [15]. The latter makes it a challenging duty to accurately detect a malicious node request; however, beyond this, the impact of a Clone ID attack can lead to serious consequences. As with any impersonation threat, the losses can involve private data sniffing, data modification and alteration, unwanted data re-transmission and, in its worst case, when the DODAG root node is compromised, the ecosystem can have repercussions from lateral movement attacks, data ex-filtration, and privilege escalation access [32].

### 2.2. Detecting a Clone ID Attack

To conduct a security evaluation of emerging technologies, such as IoT and WSNs infrastructures, the Security-in-Depth paradigm introduces the idea of minimizing any risk by maximizing the effectiveness of various security layers [33]. Within this context, security layers are the combination of physical and logical control measures that contribute to the reinforcement of abilities to prevent, detect, delay, or respond to any security disruption attempt [34]. With this, IDS, IPS, SIEM, and firewall appliances are common perimeter defensive systems, merged and placed at various network layers and operating in accordance with anomaly- and behavioural-based signatures. Once a malicious pattern is matched and flagged, the intruder may be banned or eradicated from attempting new incursions. Table 2 summarizes basic logging information gathering and detection capabilities of IoT perimeter security solutions.

Corrective controls have been proposed to achieve a certain level of prevention and mitigation towards clone node attacks over IoT and WSN protocols. In [35], the review concluded that detection and prevention techniques can fall in two main categories—centralized and distributed—for static and mobile WSNs, with different taxonomies depending on the security control algorithms and computational memory costs. It is important to highlight that, while the algorithms proposed can partially or experimentally ease some vulnerabilities, there are no ML, DL, RF, or statistical implementations to compare with. More importantly, the adjustments are not specifically oriented to RPL, thus creating a gap in the study of more intelligent solutions. Table 3 describes the proposed categories and taxonomies in the corrective controls for the mitigation of Clone ID Attacks. It is worth mentioning that the reported complexity varies exponentially, depending on the adaptations, experiments, specifications, and tests performed. This directly affects the memory cost data, as it is a measure that indicates the complexity of the number of operations that the solution must perform to observe the dynamics of each node, in order to detect a suspicious movement.

Nevertheless, it is a fact that RPL, as part of many IoT and WSN routing protocols, might have already written detection signatures for its ordinary adversarial patterns. In [32], preliminary advances reported that identity and counterfeiting attacks can be lessened by extending IDS anomaly-based signatures on specific network segments, where physical nodes establish a first authentication on the DODAG tree. In this stage, geographical information can be determined and suspicious nodes can be banned. Conversely, in [36], the authors claimed that typical IDS, IPS, firewalls, and similar defensive systems are prone to producing a high rate of false positives. Due to this, similarity comparisons are not effective for fully exploring the characteristics and taking intelligent actions on stealthy attacks, such as the Clone ID attack. A host-based IDS equipped with a ML-driven engine and a statistical learning model has been developed, in [37], in order to detect node ranking attacks on RPL, which is a crucial innovation to reduce more sophisticated attacks towards the DODAG organization. In any case, as with many newly identified IoT threats, the Clone ID attack surfaces and countermeasures in RPL networks remain unexplored.

## 3. Related Work

Little is known about how perimeter defensive systems implement persuasive mitigation rules for a Clone ID attack, or which intelligent controls by ML algorithms can classify, group, or potentially evaluate node impersonation threats [38,39] on the RPL protocol. Even so, in [40], a scheme to defeat the sybil, a closely related Clone ID attack, was proposed, which relied on radio resource testing and verification of key sets for random key pre-distribution, registration, and position verification.

In [41], various sybil attacks over large IoT networks were addressed by means of Social Graph-based Sybil Detection (SGSD), Behavioural Classification-based sybil Detection (BCSD), and Mobile Sybil Defenses (MSD). It is important to mention that such proposals may rely on preventive controls, rather than detective controls, due to an absence of analysis and monitoring at the core of RPL authentication stages.

In spite of the lack of countermeasures to detect Clone ID attacks, some authors have focused on other RPL-related attacks, specifically those affecting the DODAG node arrangement, using ML techniques. In [20], a DL-based security monitoring analysis was implemented for IoT network flows at upper layers, where a DNN algorithm was trained by means of data sets built over RPL messages with binary outcomes: Malicious and regular. To construct the predictive model, feature extraction techniques based on time windows were chosen, concluding that most ranking attacks can be substantially decreased, but covert attacks, such as the Clone ID, were on the edge to be explored in future works. Additionally, in [42], a Multi-level Perceptron (MLP) was trained with different Internet packet traces, in order to detect Distributed DoS or DoS (DDoS/DoS) attacks over RPL nodes. It is important to mention that the MLP algorithm only evaluates one type of DoS/DDoS at a time on UDP, a non-common protocol for WSNs that may not be feasible to track when counterfeited nodes propagate down the DODAG tree. Furthermore, in [43], a Deep Autoencoder Neural Network (DANN) was proposed for detecting DODAG intrusions; the algorithm was trained with remote-to-local (R2L), DoS, and user-to-root (U2R) samples from a well-known IoT data set known as NSL-KDD [44]. Even though Autoencoders can maximize the way that a DANN projects network features, the unbalanced nature of the NSL-KDD data set caused TCP/IP packets to compose the majority of samples, overshadowing the minority of RPL packets adjacent to a DODAG; hence, the RPL protocol was out of scope of their paper. In a more conscientious work [45], a DNN in consolidation with a Self-Taught Learning (STL) framework was developed, with two main tasks to cluster different RPL attacks: Feature learning and dimensional reduction, with different Autoencoder configurations. The results revealed that the STL can group attacks into four categories with an improved feature analysis: DoS, Probe, R2L, and U2R attacks. Notwithstanding its recommendation statuses, impersonation remained undetected. Although former approaches have tackled some RPL attacks on various DODAG fields using ML, the Clone ID attack has been noted as warranting subsequent exploration. Table 4 summarizes the algorithms and data acquisition from the previously discussed work.

## 4. Proposed Framework

The workflow of the proposed methodology is depicted in Figure 5. First, in a pre-processing stage, the data set is subjected to class-balancing procedures, value transformations, and scaling. Afterwards, unsupervised learning is performed, in order to find the most adequate features to represent a Clone ID Attack from RPL messages. Then, supervised learning is used to classify samples into one of two main classes: Attack or normal. Finally, the resulting model is evaluated in terms of the following performance metrics: Accuracy, precision, and F1 score.

### 4.1. Data Collection

To extract Clone ID attacks on RPL network conversations, a virtual environment was deployed using Cooja [46], a well-known WSN emulator, and tshark, a network packet analyzer [47]. It is important to mention that the capabilities of Cooja allow for emulating traffic and working with real software for attack simulation. According to an exhaustive comparison with other IoT simulators, presented in [48], this simulator allows for the estimation of the effects of an attack, the consumption of hardware resources, and the support for different operating systems. The only disadvantage is the high consumption of resources in minimal environments. The scenario was configured on a 6LowPAN network over RPL messages, involving o node built using a Zoleria Z1 device, 100 Z1-based dummy sensors spread across a radius of 10 and 200 m, and ten malicious sensors with Z1 embedded technology. Three types of topological structures were designed to resemble real traffic, with impersonated nodes capable of being dumped from network captures, to build the three final data sets. The first data set consisted of twenty nodes with two malicious sensors; the second one was comprised of fifty nodes with five malicious nodes; and, ultimately, the third data set contained one-hundred nodes, of which ten were malicious. Each topology had a root node or sink, which was assumed to be exposed to an external network. In Table 5, the topologies fashioned to mirror real Clone ID attacks are listed.

The IoT virtual environment to exploit the Clone ID attack, carried out using the Cooja simulator, is depicted in Figure 6.

The topology, node distribution across established radius distances, and configuration of each data set are illustrated in Figure 7a–c.

As with any other raw network traffic examination, samples may contain many headers whose values are optional or null, thus representing a missing data problem. For this reason, instead of fulfilling a high-dimensional data representation by statistical means, incomplete headers were filtered and discarded. After replaying live network captures, a total of 1207 headers were identified as candidate features, but only 19 field headers were selected with full and complete information. Additionally, it should be noted that some features were listed as categorical values—*wpan.ack_request, wpan.pending, icmpv6.code, wpan.dst_addr_mode, and 6lowpan.pattern*—due to their ordinal behaviour. Therefore, datatype casting was carried out to transform them into numerical types. To construct a tabular data set, each row was a sample. Columns represent features, according to the previously selected numerical field headers. The last column maps samples to one of two class labels, where 0 corresponds to a *Normal* network conversation (i.e., packets sent from a legitimate node to another legitimate node) and 1 corresponds to a *cloned ID attack*. Table 6 delineates the features selected after RPL network flows were monitored and recorded.

### 4.2. Data Pre-Processing

In this subsection, the pre-processing steps aimed at cleaning and preparing the data set for posterior phases, such as training and model evaluation, are detailed. In a first data exploration, categorical values were codified or transformed into discrete numerical sequences. Secondly, characteristics that contained outliers that affect the statistical variance and sparseness of the whole data set were scaled and standardized. Finally, due to the unbalanced nature of class samples, a data compensation technique to undersample the class distribution was applied. The sequence of steps used to process each data set is listed in Algorithm 1, and described as follows:*Data set balancing*: As stated in [49], the samples within network captures are considerably smaller than those from benign applications, leading to the possibility of overfitting and classification downgrading. That being the case, algorithm estimations may always generalize the majority class features, overlapping the minority ones [50]; for example, in [51] the importance of data set balancing regarding a cervical cancer prediction model (CCPM) using risk factors as inputs was emphasized. In this case, the authors balanced their data set by using a synthetic minority over-sampling technique (SMOTE), due to their use of a Random Forest classifier. Although SMOTE performs better than other re-sampling techniques in traditional machine learning scenarios, in accordance with [52], Random Over-Sampling methods are better, as, in a real network traffic detection and filtering scenario, the generation of SMOTE samples could not be practical for high dimensional data. It would also not be ideal to use under-sampling, as it has been shown that, by removing samples from a majority class, key evidence that may be useful in feature engineering procedures could be lost. Therefore in the pre-processing stage in this proposal, a Random Over-Sampling (ROS) procedure was performed with no replacement.*Value transformation*: Features that contain nominal and categorical data, such as IPv6 source and destination addresses (ipv6_src and ipv6_dst), were transformed into discrete values (Label Encoding), in a range between 0 (the first IPv6 host) and *n* (the last one). In the case that a feature was comprised of a categorical sequence (i.e., {ipv6_dst, ipv6_src, icmpv6_code, wpan_dst_addr_mode, wpan_fcf, and sixlowpan_pattern}), the transformation was a reduction to a sparse numerical array using One-Hot Encoding (OHE). After the conversion, each series were replaced by values between 0 (representing the absence of addresses) and 1 (active values).*Scaling*: Numerical features were standardized to guarantee equal weights during the learning process [50]. Specifically, standard scaling was used on each numerical feature, x∈X, to center its mean, μ=0, and scale it with respect to the standard deviation σ, as shown in Equation (Equation 1).
(1)$$x_{standardized}=\frac{x-\mu }{\sigma }.$$


**Algorithm 1:** Preprocessing tasks over each data set.

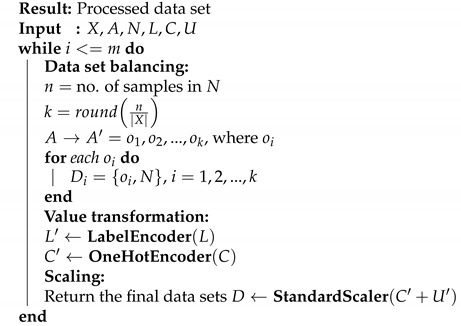



To interpret the steps involved in the processing algorithm, the following definitions are denoted:$x_{i=1}^{m}\in X$ is the set of original samples, where *x* is a sample and *X* is the data set;A⊂X,N⊂X;∀|A|<|N| are the subsets belonging to minority and majority classes, respectively;*L = {ipv6.src, ipv6.dst}* represents the set of categorical features to encode to discrete values;*C = L ∪ {icmpv6.code, wpan.dst_addr_mode, wpan.fcf, 6lowpan.pattern}* represents the set of categorical features to transform to real values;*U = {frame.time_delta, frame.time_epoch, frame.time_relative, frame.cap_len, frame.len, frame.number, wpan.fcs, wpan.frame_length, wpan.seq_no, ipv6.plen, icmpv6.checksum}* represents the set of numerical features;*k* is the number of minority samples *A* contained in *X*; and{cloneid_20n,cloneid_50n,cloneid_100n}∈D is the final balanced set, resulting from Label Encoding, OHE, and random selection and sampling without replacement.

In Table 7, the data sets after the pre-processing steps are detailed.

It is worth mentioning that, for the *cloneid_50* and *cloneid_100* data sets, the number of features increased as a result of the encoding steps. While more nodes were transformed, the sequences of IP addresses encoded by OHE required more characterization.

### 4.3. Unsupervised Pre-Training

The topologies contained in each data set *D* conformed to IoT network packets, which can vary from different origins, devices, latency, and changing behaviours, that depend directly on the environment that is being observed. Although data exploration, transformation, and scaling are mandatory steps to construct proper inputs for the learning algorithm, feature engineering is a key step to select relevant information, reduce the computational costs from high-dimensional samples, and filter noisy data; inherent factors from IoT–RPL communications [53]. To reach adequate feature extraction and selection, in this work, we propose the use of an unsupervised learning algorithm, known as Autoencoder [54,55], a data representation method capable of tracking network packet variations by backpropagation and reconstructing the output values as being equal to the initial inputs using an identity function; in other words, with minimum reconstruction error. A simple Autoencoder is depicted in Figure 8.

The structure of an Autoencoder is composed of an input layer *Layer $L_{1}$*, a hidden layer *Layer $L_{2}$*, and an output layer *Layer $L_{3}$*. For the most part, the hidden units in *Layer $L_{2}$* learn from a lower-dimensional representation of the input, as this layer has fewer units [56]. In the encoding phase, the hidden representation is compacted to a vector $h_{d}$, mapped with an input vector $x_{d}$ containing *d* features:(2)$$h_{d}=f_{\theta \left(x_{d}\right)=\sigma \left(Wx_{d}+b\right)},$$
where $f_{\theta }$ is the mapping function, *W* is the p×p weight matrix for the *p* hidden units, *b* is the bias vector, θ is the mapping parameter set θ=W,b, and σ is the sigmoid activation function.

At the same time, in the decoding phase, the reconstructed d−dimensional vector $y_{n}$ is computed by mapping back the condensed hidden representation $g_{\theta }$ of vector $h_{n}$, as follows:(3)$$y_{i}=g_{\theta \left(h_{n}\right)=\sigma \left(Wh_{n}+b\right)}.$$

Finally, to minimize the reconstruction error between input and output for *m* samples, the following model is computed:(4)$$E\left(x,y\right)=\frac{1}{m}∥\sum_{i=1}^{m}\left(x_{n}-y_{n}\right){∥}^{2}.$$

To find the latent representation of each data set *D*, a vector $h_{d}$ was used as a feature selector, through an $\ell_{1}$ regularization task implementing a sparsity constraint (also called Sparse Auto Encoding; SAE), for the hidden layer. This regularization shrinks noisy feature weights towards zero and extracts the most relevant ones. The Autoencoder and SAE configurations, activation functions, epochs, and batch sizes are depicted in Figure 9.

The configuration employed two-thirds of the total number of input neurons over the hidden layer; the Rectified Linear Unit (ReLU) activation function was used to simplify training and improve performance [57]; Adam was utilized as an optimizer, with a gradient descent algorithm to adapt the learning rate [49]; and $\ell_{1}$ regularization with a coefficient of 0.0001 was established [58].

### 4.4. Supervised Classification

The growing popularity of AI-based solutions to identify and classify malicious behaviors in different IoT networks and protocols has led to a vast range of schemes supporting supervised classification algorithms. To build a custom AI solution to detect a Clone ID Attack, the IoT surface and the previously pre-processed features must be in accordance with the natural capabilities of a chosen algorithm. In [59], counterfeiting attacks were cataloged as Network Service Surface Threats, encompassing Network Layer defenses and Effective IoT Security Controls, with dynamical and temporal features. With this preamble, the scattered and noisy form of Clone ID attack network captures may be out of the spectrum of well-known shallow algorithms, including Decision Trees (DT) [60], Support Vector Machines (SVM) [61], Naive Bayes (NB) [62], K-Nearest Neighbours (KNN) [63], and Association Rules (AR) [64]. In Table 8, some flaws detected in classic algorithms towards IoT attack and threat recognition are presented.

Nonetheless, DL algorithms [65] have an advantage over shallow algorithms when dealing with more complex data representations of IoT network flows, by thoroughly inspecting the features in various hidden layers. To deeply contextualize the Clone ID Attack, a Dense Neural Network (DNN) architecture [66] was modeled to strengthen feature propagation and maximize the recognition of network conversations with impersonated nodes.

In a DNN, each neuron receives a weighted sum of the outputs of the neurons connected to them, making faster computations to learn estimations over the training sets of *D*. The model was adapted with two hidden layers and an output layer consisting of one unit, as shown in Figure 10.

To minimize the classification error, a Binary Cross-Entropy Loss (BCEL) activation function was added, as follows:(5)$$J\left(W,b;x,y\right)=-(y\log\left(h_{W,b}\left(x\right)\right)+\left(1-y\right)\log\left(1-h_{W,b}\left(x\right)\right)),$$
where $\left(x^{\left(i\right)},y^{\left(i\right)}\right)$ are the samples of the training set $X_{train}\in D$, $h_{W,b}\left(x\right)$ is the hypothesis equation with parameters *W* (weight) and *b* (bias), and *y* is the ground truth (the true label). Moreover, the DNN configuration and hyper-parameters used in this architecture are depicted in Figure 11.

## 5. Results and Discussion

This section presents and discusses the experimental results obtained from the Unsupervised pre-training and Supervised classification. To configure the learning steps, the sets {cloneid_20n, cloneid_50n, cloneid_100n}∈D were split into a training subset $X_{train}$ built with 80% of random samples from each data set contained in *D*, a validation set $X_{val}$ with 20% of random samples from $X_{train}$, and a test set $X_{test}$ containing 20% of random samples of each data set in *D*. In total, three different approaches were proposed for the conjunction of data pre-processing, Unsupervised pre-training, and Supervised classification stages, as detailed in Table 9.

To evaluate the performance of the proposed models, two metrics were calculated: *Accuracy* and *F1 score*. The accuracy is an indicator of the total number of correct predictions made by the resulting classification model:(6)$$Accuracy=\frac{T_{P}+T_{N}}{T_{P}+T_{N}+F_{P}+F_{N}}$$

Equation (Equation 6) is based on the premise of True Positives ($T_{P}$), True Negatives ($T_{N}$), False Positives ($F_{P}$), and False Negatives ($F_{N}$), where:
$T_{P}$ is the number of attacks classified as attacks;$T_{N}$ is the number of normal conversations classified as normal;$F_{P}$ is the number of normal conversations misclassified as attacks; and$F_{N}$ is the number of attacks misclassified as normal conversations.

The *F1 score* is determined as the harmonic mean of precision and recall (i.e., as a compensation between $F_{P}$ and $F_{N}$), thus describing whether the resulting classification model performs as expected:(7)$$F1-score=2\times \frac{Precision\times Recall}{Precision+Recall},$$
where *Precision* is the mathematical baseline between the number of positive predictions and the total number of positive class values predicted (i.e., how precise the positive predictions are):(8)$$Precision=\frac{T_{P}}{T_{P}+F_{P}},$$
and
(9)$$Recall=\frac{T_{P}}{T_{P}+F_{N}}$$

In an independent manner, a test was conducted using *k*-fold cross-validation, taking into account the time span since the training procedure started and finished. Table 10, Table 11 and Table 12 depict, in bold rows, the best *Precision* and *F1-scores* values for {cloneid_20n, cloneid_50n, cloneid_100n}∈D sets, subjected to the proposed models configurations.

As the results show, the most effective models were the DNN+SAE configurations for the cloneid_50n and cloneid_100n data sets. Even though the cloneid_20n data set presented the lowest performance metrics, the implementation of SAE and AE Autoencoders were capable of finding a latent representation for classification inputs, compared with non-encoded data, and improved the classification results. The time spans monitored on the training steps changed significantly when applying SAE + DNN on cloneid_50n and cloneid_100n, proving that Autoencoders suit characterization tasks to further SL inputs. As stated in Section 1, this is the first work to build a framework to detect Clone ID attacks in the RPL protocol, aiming to enable comprehensive data exploration, a feature extraction procedure, and a training configuration, with the proper representation of more than one million samples from an IoT environment. Even though there exists little literature to deliver a deeper comparison with other state-of-the-art approaches, the survey described in Section 3, Table 4, was considered, in order to compare the proposed framework with related works that classified/detected security threats in IoT environments using ML techniques. Table 13 summarizes the works selected for comparison.

As observed in Table 13, the SAE + DNN classifier used to build the proposed framework performed better, on average, than the DFFN, MLP, A-DNN, and SAE + SVM. Even though Yavuz [20] employed additional features related to rates and packet counts, their results showed that only 19 features were sufficient to train a DFFN, which can hardly represent actual IoT network traffic; therefore, a Clone ID Attack may be undetectable. Rezvy et al. [43] and Al-Qatf et al. [45] achieved similar results compared to this proposal; however, the features were extracted from a pre-built data set which was limited to TCP/UDP traffic, mostly from non-IoT messages. As a future work, deploying the SAE+DNN as a whole framework on a IDS/IPS module is suggested; see the mechanism illustrated in Figure 12.

As reported in [68], the integration of a real-time AI perimeter security for IoT threat detection must contain the following workflow: 1. A physical or virtual network interface must be listening for traffic coming from a WSN, where specific RPL protocol messages must be filtered and sent to feed a detection agent; 2. The detection agent is responsible for analysing, training, and predicting malicious traffic observations from filtered RPL data packages. This block consists of two main modules: (a) A Baseline modelling, aimed to split data into time windows, in order to accurately model complex traffic observations, as well as a data pre-processing stage to balance, scale, and standardize the previously gathered data and improve the data set that will be further subjected to feature extraction and training tasks using the SAE + DNN algorithms. (b) Prediction: With the final data set trained by SAE + DNN, a predictive model can be compiled to classify the traffic as benign or malicious. If the sample was classified as malicious, the response and mitigation rules shall perform the corresponding actions, depending on the impact of the threat. Although the implementation is useful, in comparison with corrective controls and shallow algorithms, it is important to contrast that, for instance, DL approaches work with batch learning, which is designed to perform only one training stage and produce one predictive model. Still, as proposed in [69], one of the challenges in AI-based intrusion detection systems is predictive model updating; therefore, online learning should be adapted, where new traffic patterns can be analyzed in real-time by adapting the DNN weights as incremental training is performed. This could be difficult to achieve by an IDS/IPS, due to its physical memory and processing limitations, as AI optimization functions—specifically DL algorithms—are not convex and require more resources to run their processes. An alternative solution is cloud processing, where elastic resources could grow on demand and perform the training tasks, returning an updated predictive model through an API request. In addition, a more in-depth feature analysis techniques can be employed which, according to [70], can fuse information from different sensors, extending the capabilities of the framework to other protocols and structures, taking into account feature fusion, attribute selection, and feature weighting tasks.

## 6. Conclusions

With the extensive use of IoT technology in critical infrastructures, it is crucial to address an in-depth security approach, where a detective layer must be put in place. In this paper, we presented one of the first frameworks that employs ML to tackle a critical and stealthy threat; namely, the Clone ID Attack on the RPL protocol. Two solutions can minimize the impact of a Clone ID attack: Corrective and device-based perimeter defensive replication attacks. Centralized or distributed corrective controls in WSNs make use of hardware changes, software directives, the use of base stations to monitor node changes and authorizations, and cryptographic configurations; however, the cost of design and implementation is quite expensive, considering the need for stability in minimal environments such as IoT. Furthermore, the scalability of such systems is almost null, as stealthy attacks can become more sophisticated, making previous corrections obsolete. Perimeter defense systems (IDS/IPS) can generate a high false positive rate, as they require behavioural signatures to make a similarity match, which is a high-cost technique and faces a race over time in generating new signatures to address sophisticated threats such as the Clone ID Attack. Therefore, it is imperative to have intelligent mechanisms that can adapt the knowledge of new samples from different behavioral characteristics and detect with different levels of performance any abnormal pattern. To construct the framework, an IoT network emulator was employed to simulate counterfeiting attacks with three different topologies: Twenty sensors with two malicious nodes, fifty sensors with five malicious nodes, and one hundred sensors with ten malicious nodes. After capturing and labeling the network traffic, a thorough data exploration was conducted, in order to transform, label, and encode values to a proper numerical representation. Then, two types of Autoencoders (SAE and AE) were proposed as feature selectors and extractors, in order to feed a Dense Neural Network (DNN). Although there are no existing works that apply ML techniques to detect, classify or cluster Clone ID Attacks, the SAE+DDN architecture proved that, by comparing with other ML and DL methods, the performance metrics can a can achieve an accuracy of 99.65%. As future work, the affordable deployment of the AI-based module, on IPD/IPS perimeter security systems with a detection agent, can be integrated with filtering capabilities to work in conjunction with a compiled classification model. Further exploration of other less widely researched RPL attacks, such as Sinkhole, Blackhole, or Selective Forwarding, will be added, in order to test the proposed framework and compare and analyze whether it is capable of detecting a wider range of stealthy menaces. Besides, future research can be conducted in other standards and broadly used IoT protocols in the application layer, such as MQTT and CoAP.

## Figures and Tables

**Figure 1 sensors-21-03173-f001:**
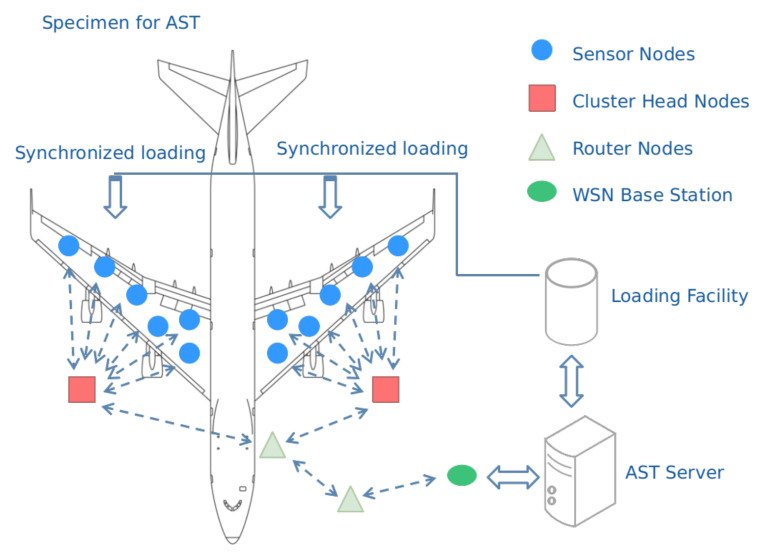
The AST–WSN topology comprises cluster head nodes that forward data to the router nodes, which finally sends the sensed information to an AST Server for storage and analysis.

**Figure 2 sensors-21-03173-f002:**
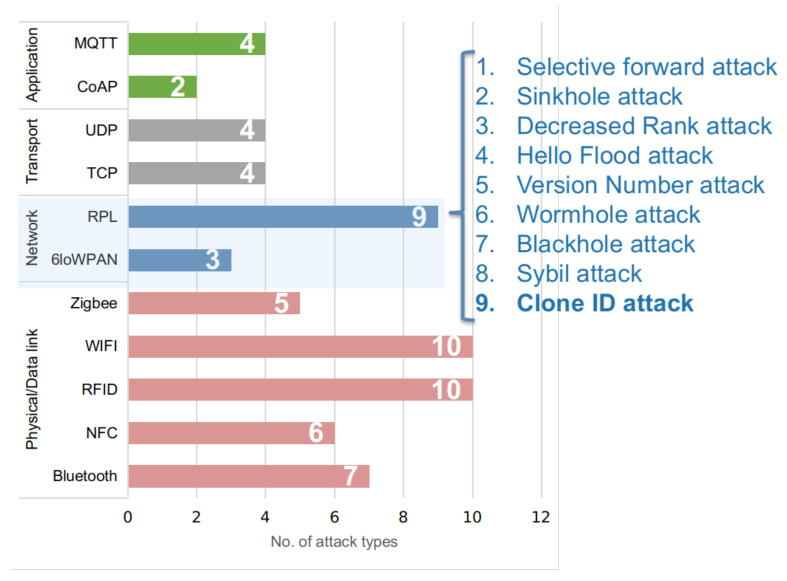
Number of cyber-attack types directed at IoT protocols, emphasizing those aimed at RPL and LLNs.

**Figure 3 sensors-21-03173-f003:**
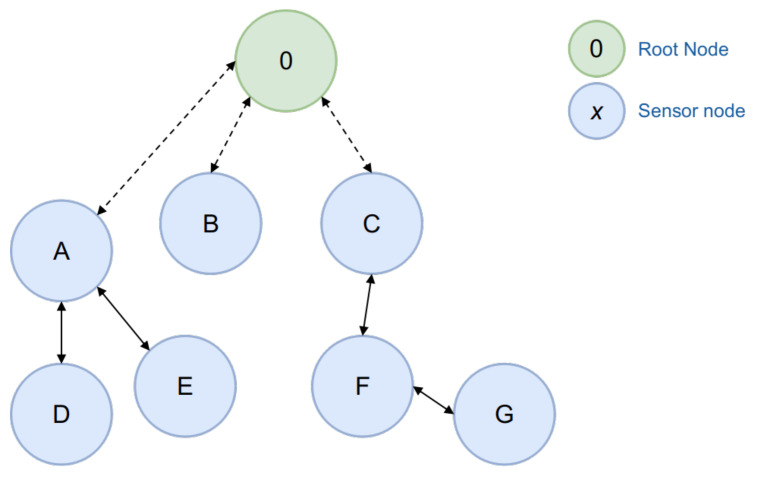
Concept diagram of DODAG node organization.

**Figure 4 sensors-21-03173-f004:**
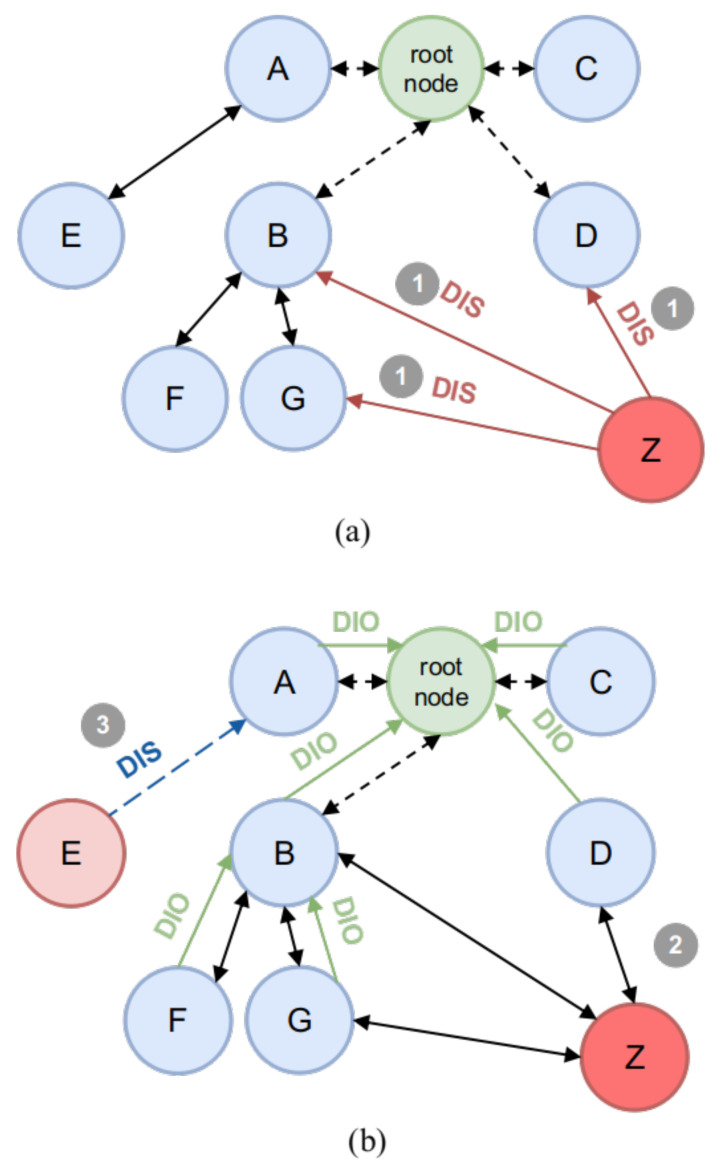
(**a**) Clone ID attack launching and (**b**) outcome.

**Figure 5 sensors-21-03173-f005:**
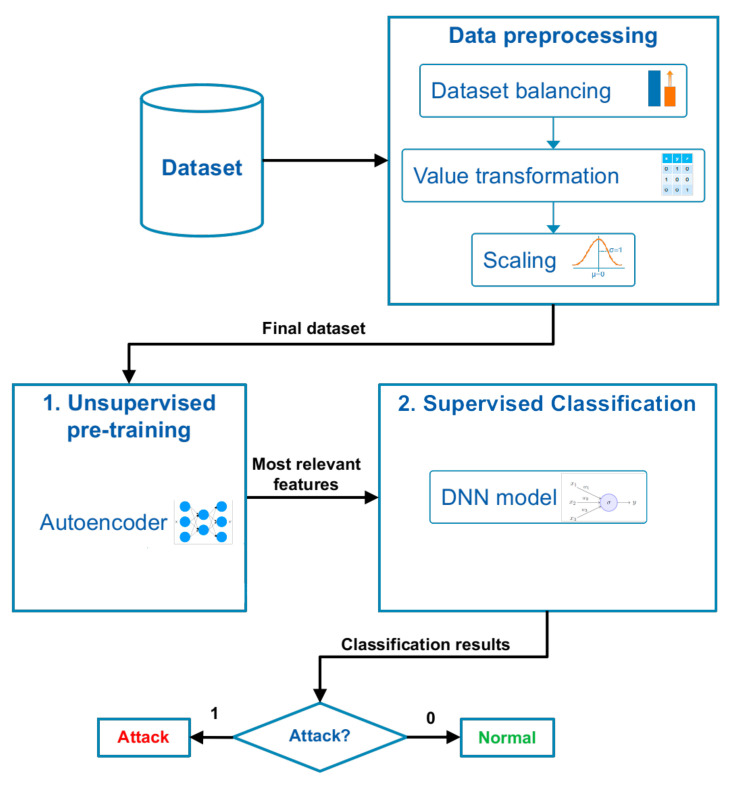
Workflow of the proposed methodology.

**Figure 6 sensors-21-03173-f006:**
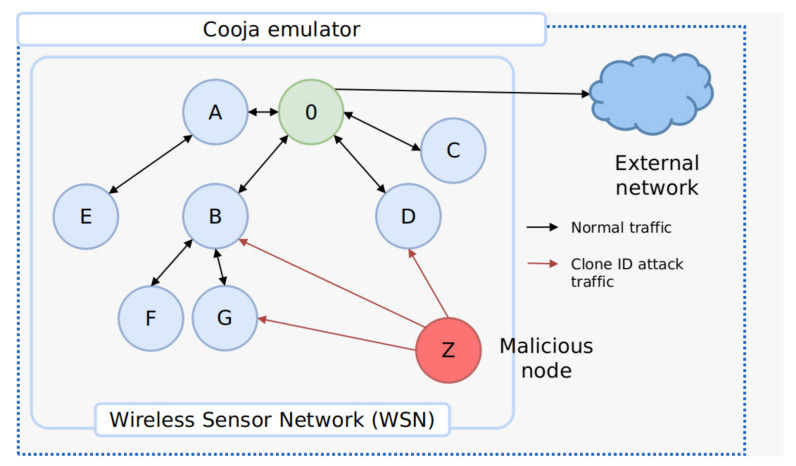
IoT virtual environment to simulate the Clone ID attack.

**Figure 7 sensors-21-03173-f007:**
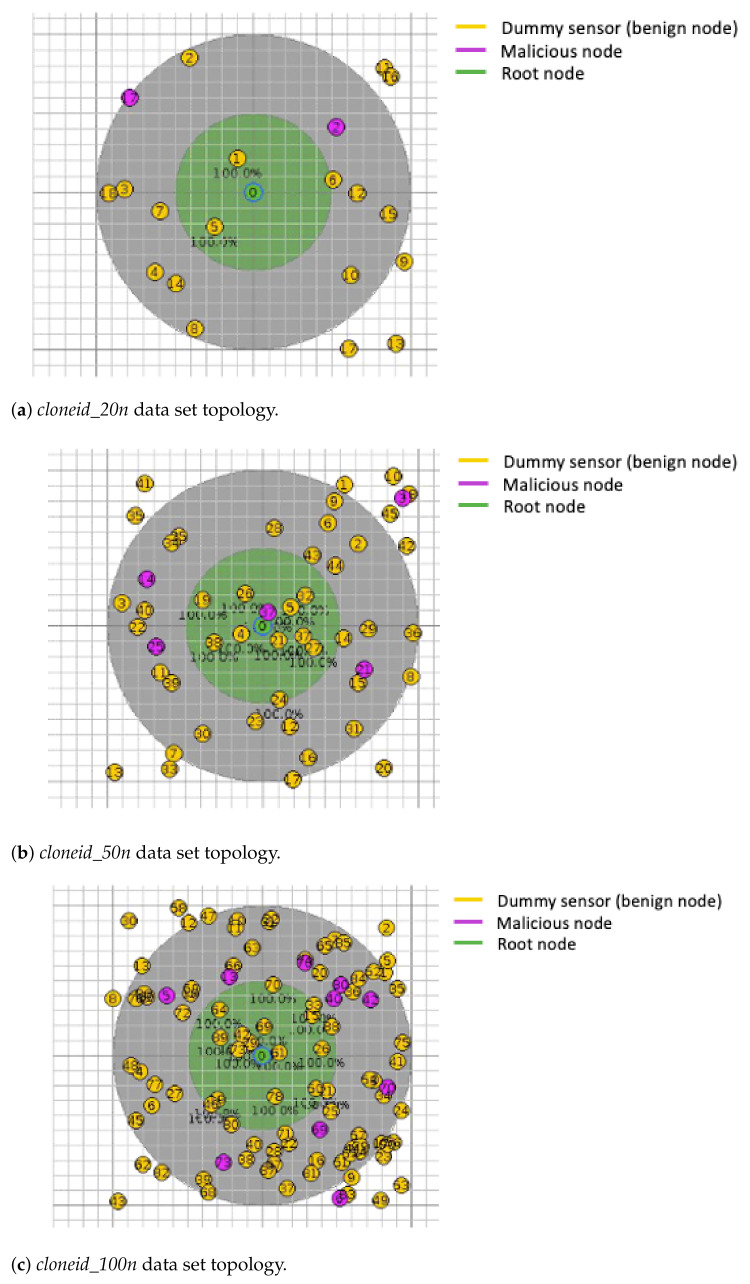
Sensors employed to simulate a Clone ID attack, with different topologies and node configurations.

**Figure 8 sensors-21-03173-f008:**
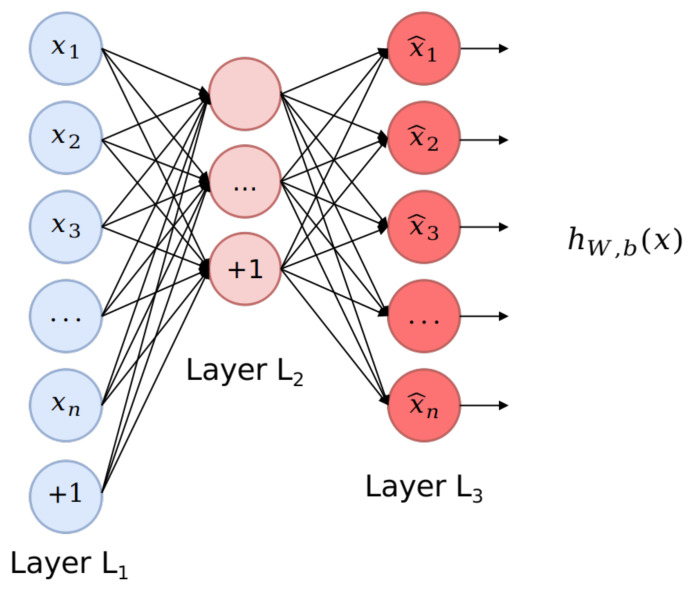
Conceptual diagram of an Autoencoder.

**Figure 9 sensors-21-03173-f009:**
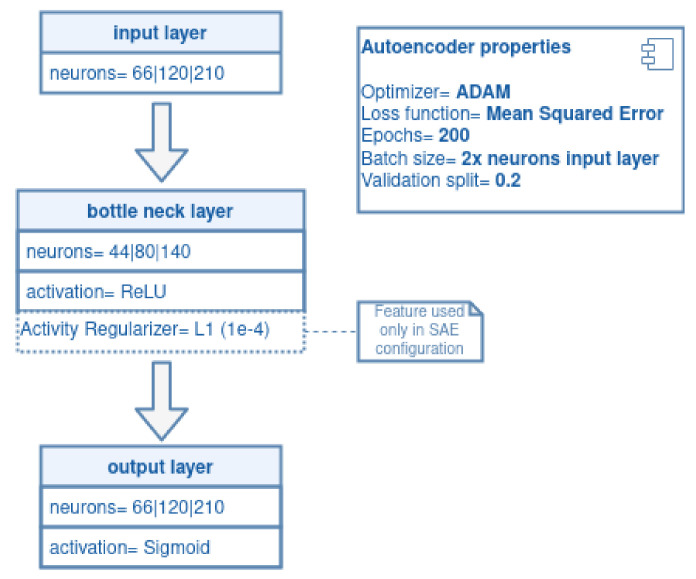
Autoencoder and SAE configurations.

**Figure 10 sensors-21-03173-f010:**
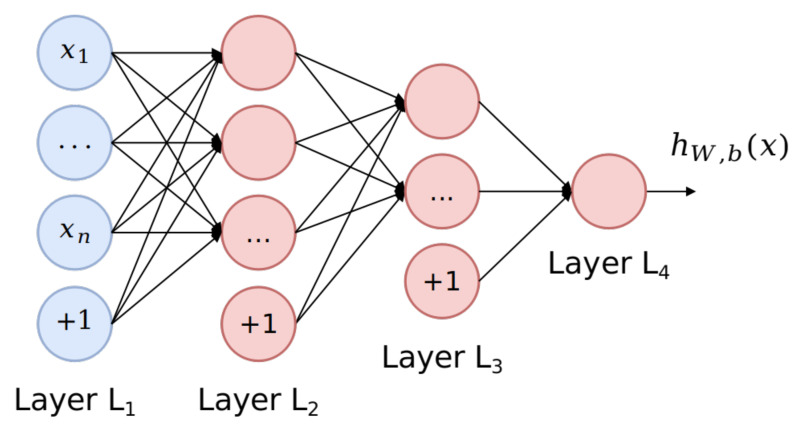
Architecture of the DNN.

**Figure 11 sensors-21-03173-f011:**
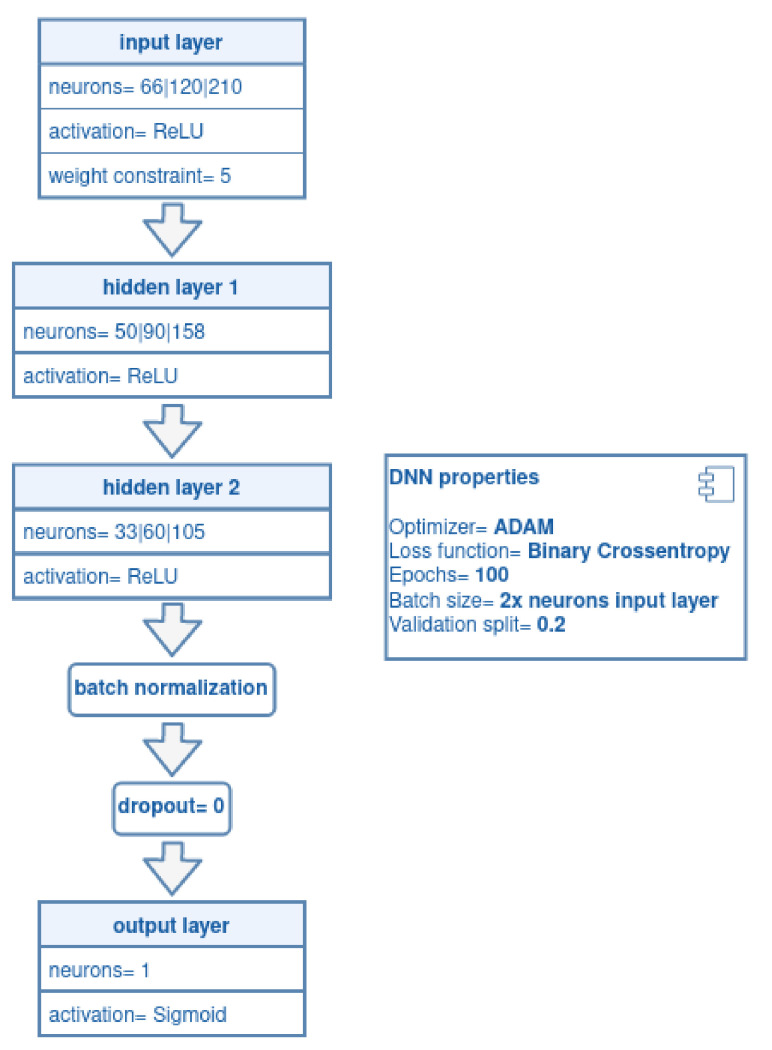
DNN configuration and hyper-parameters.

**Figure 12 sensors-21-03173-f012:**
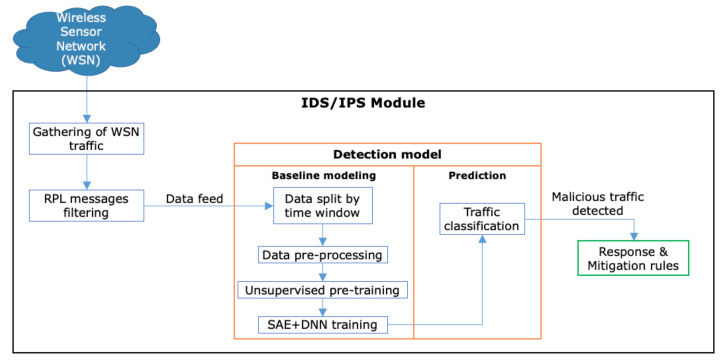
Proposed module integration for an IDS/IPS using the proposed framework.

**Table 1 sensors-21-03173-t001:** Codes for RPL control message types.

Code Field ID	Description
0x00 0x01 0x02 0x03 0x80 0x81 0x82 0x83 0x8A	DODAG Information Solicitation (DIS) DODAG Information Object (DIO) Destination Advertisement Object (DAO) Destination Advertisement Object Acknowledgment (DAO-ACK) Secure DODAG Information Solicitation Secure DODAG Information Object Secure Destination Advertisement Object Secure Destination Advertisement Object Acknowledgment Consistency Check

**Table 2 sensors-21-03173-t002:** Common logging information gathering and detection capabilities of perimeter security solutions. Those who interact with IoT protocols are highlighted in bold.

Logging Information Gathering Capabilities	Detection Capabilities
-Timestamp (e.g., date and time)	-Application layer reconnaissance and attacks
-Connection or session ID	-**Network layer reconnaissance and attacks**:
-Event or alert type	***Sinkhole attack*** ${}^{‡}$;
-Rating (e.g., priority, severity, impact, confidence)	***Neighbour attack;*** ${}^{‡}$
-**Network, transport, and application layer IoT protocols:**	***DIS attack*** ${}^{‡}$ and
*CORPL* ${}^{†}$; *CARP* ${}^{†}$; *6LoWPAN* ${}^{†}$ and ***RPL*** ${}^{†}$	***Local repair attack*** ${}^{‡}$
-Source and destination IP addresses	-Unexpected application services
-Source and destination TCP or UDP ports, or ICMP types and codes	
-Number of bytes transmitted over the connection	
-Decoded payload data, such as application requests and responses	
-State-related information (e.g., authenticated username)	

^†^ IoT network protocols; ^‡^ already recognized RPL-oriented attacks.

**Table 3 sensors-21-03173-t003:** Centralized and distributed techniques for corrective controls in the detection of Clone ID Attacks.

Category	Description	Taxonomies	Memory Complexities Reported *
Centralized	Uses a powerful central Base Station (BS) to track each node position and its neighbours identity when joining to the network	Key usage-based Base station-based Neighbourhood social signature-based Cluster head-based Zone-based Neighbour ID-based	$O\left(d\right),O\left(d\sqrt{n}\right),O\left(log\left(n\right)\right),O\left(1\right),O\left(N\right)$ $O\left(d\right)+\min\left(Mwlog_{2}M\right)$ O(n) O(t) $\frac{O(d)}{O(nZ)}$ O(n)
Distributed	Clone replication is applied to all network nodes with no central Base Station (BS)	Node to network broadcasting Witness node-based Generation- or group-based Neighbour-based Clustered-based Whiteness path-based Cluster head-based	O(1) $O\left(g\right),O\left(d\right),O\left(\sqrt{n}\right),O\left(\omega \right)O\left(tk+t^{\prime }k\sqrt{n}\right),O\left(t+t^{\prime }{\sqrt{n}}^{\prime }\right)$ $O\left(\sqrt{n}\log\left(n\right)\right),O{\left(1^{2}\right)}^{2},O\left(h\right),O{\left(1\right)}^{2}$ $O\left(1\right),O\left(m\right),O\left(d+2m\right),O\left(2+2xm\left(1+D_{max}\right)\right),O\left(r,\sqrt{n}\right),O\left(r\right)$ O(r) O(k,e) $O(\frac{l}{r})$ $O(\frac{1}{p})$

^*^ results expressed in BIG-O notation.

**Table 4 sensors-21-03173-t004:** Related work using ML models on RPL and DODAG attacks.

Authors	ML Algorithm	Attack	Data Set
Yavuz, F. Y. et al. [20]	Deep Feed-Forward Network (DFFN)	Decreased rank	Custom WSN data
		Hello flood	
		Version number	
Hodo et al. [42]	MLP	UDP DDoS/DOS	NSL-KDD
Al-Qatf et al. [45]	SAE and SVM	DoS, Probe, R2L, U2R	Custom TCP/UDP traffic

**Table 5 sensors-21-03173-t005:** Data set sizes and corresponding topologies.

Data Set Name	No. of Nodes	Malicious Nodes	Benign Nodes	Samples
*cloneid_20n*	20	2	18	1,232,862
*cloneid_50n*	50	5	45	1,576,668
*cloneid_100n*	100	10	90	1,492,579

**Table 6 sensors-21-03173-t006:** Feature descriptions for the data set.

No.	Field Name	Description	Type of Feature
1	frame.cap_len	Frame length stored into the capture file	Numerical
2	frame.len	Frame length on the wire	Numerical
3	frame.number	Frame Number	Numerical
4	frame.time_delta	Time delta from previous captured frame	Numerical
5	frame.time_epoch	Epoch Time	Numerical
6	frame.time_relative	Time since reference or first frame	Numerical
7	wpan.ack_request	Acknowledge Request	Categorical
8	wpan.dst_addr_mode	Destination Addressing Mode	Categorical
9	wpan.fcf	Frame Control Field	Numerical
10	wpan.fcs	Frame Check Sequence	Numerical
11	wpan.frame_length	Frame Length	Numerical
12	wpan.pending	Frame Pending	Categorical
13	wpan.seq_no	Sequence Number	Numerical
14	6lowpan.pattern	Pattern	Categorical
15	ipv6.dst	Destination	Categorical
16	ipv6.plen	Payload Length	Numerical
17	ipv6.src	Source	Categorical
18	icmpv6.checksum	Checksum	Numerical
19	icmpv6.code	Code	Categorical
20	class	Normal or attack class	Numerical

**Table 7 sensors-21-03173-t007:** Data set descriptions after the pre-processing steps.

Data Set Name	No. of Features	Samples
*cloneid_20n*	67	1,749,976
*cloneid_50n*	121	2,131,328
*cloneid_100n*	211	2,078,832

**Table 8 sensors-21-03173-t008:** Comparison of different shallow algorithms and detected flaws.

Algorithm	Drawbacks for Detecting IoT Attacks and Threats
DT [60]	Large data storage, computational complexity with high-dimensional network features, prone to over-fitting
SVM [61]	Overlapping of class samples with large data sets, such as IoT network samples
NB [62]	Inaccurate for finding feature relationships in complex data representations, comparable to impersonation and sybil attacks
KNN [63]	Flawed and time-consuming processes for finding optimal neighbours over raw data corresponding to IoT packets
AR [64]	Ineffective to map efficient rules in large IoT network nodes

**Table 9 sensors-21-03173-t009:** Ensemble of Autoencoders and DNN architectures.

No. of Model	Configuration
1	No Autoencoder + DNN
2	SAE + DNN
3	AE + DNN

**Table 10 sensors-21-03173-t010:** Performance metrics for the cloneid_20n data set.

No. of Model	Configuration	Accuracy	F1-Score	Total Time	Complexity *
**1**	**SAE + DNN**	**96.72**	**96.70**	**3:29:44**	$\Omega (2^{h})$
2	AE + DNN	94.41	94.43	3:13:54	$\Omega (2^{h})$
3	No Autoencoder + DNN	93.46	93.36	4:20:30	$\Omega (2^{h})$

^*^ Based on the proposition described in [67], where it was explained that the hidden units of deep networks can grow exponentially, where *h* is the number of hidden units and Ω specifies that the algorithm will at least take a certain amount of time to produce and operate, without exceeding a certain period of time.

**Table 11 sensors-21-03173-t011:** Performance metrics for the cloneid_50n node data set.

No. of Model	Configuration	Accuracy	F1-Score	Total Time	Complexity *
**1**	**SAE + DNN**	**99.65**	**99.65**	**2:56:20**	$\Omega (2^{h})$
2	AE + DNN	99.08	99.08	4:05:47	$\Omega (2^{h})$
3	No Autoencoder + DNN	99.04	99.04	3:16:44	$\Omega (2^{h})$

^*^ Based on the proposition described in [67], where it was explained that the hidden units of deep networks can grow exponentially, where *h* is the number of hidden units and Ω specifies that the algorithm will at least take a certain amount of time to produce and operate, without exceeding a certain period of time.

**Table 12 sensors-21-03173-t012:** Performance metrics for the cloneid_100n node data set.

No. of Model	Configuration	Accuracy	F1-Score	Total Time	Complexity *
**1**	**SAE + DNN**	**99.25**	**99.26**	**1:40:48**	$\Omega (2^{h})$
2	AE + DNN	98.66	98.66	2:19:50	$\Omega (2^{h})$
3	No Autoencoder + DNN	98.53	98.53	1:41:24	$\Omega (2^{h})$

^*^ Based on the proposition described in [67], where it was explained that the hidden units of deep networks can grow exponentially, where *h* is the number of hidden units and Ω specifies that the algorithm will at least take a certain amount of time to produce and operate, without exceeding a certain period of time.

**Table 13 sensors-21-03173-t013:** Comparison with well-known works that presented ML approaches to classify security threats in IoT environments.

Author	Algorithm	Accuracy
Yavuz, F. Y. [20]	Deep Feed Forward Network (DFFN)	94.9%
Hodo et al. [42]	Multi-level perceptron (MLP)	99.4%
Rezvy et al. [43]	Autoencoder A-DNN (DNN)	99.3%
Al-Qatf et al. [45]	SAE+SVM	99.4%
This proposal (cloneid_20n data set)	SAE + DNN	96.72%
**This proposal (cloneid_50n data set)**	**SAE + DNN**	**99.65%**
This proposal (cloneid_100n data set)	SAE + DNN	99.25%

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
