# Peer review of "A Dense Neural Network Approach for Detecting Clone ID Attacks on the RPL Protocol of the IoT"

_sensors, 2021, doi:10.3390/s21093173_

Round 1

Reviewer 1 Report

1. There are many misspellings in this article, which the author should correct. For example, in line 41, the word "asses" should be corrected to "assess". In Figure 12, the authors should explain why "ANN" is used instead of "DNN", because the authors use DNN for classification instead of ANN, and what is the meaning of FN' in Equation 6. The authors state in Table 4 that the malicious nodes of cloneid_50n are five, but in lines 448 and 449, it is stated that fifty sensors with two malicious nodes.

2. Figure 2 shows that there are 9 kinds of attacks in RPL. The authors need to explain how high the possibility of clone ID attack is and explain why they only discuss the motive and reason of clone ID attack. In Table 12, the authors should compare with other methods of classifying clone ID attacks and should not compare with other methods of classifying attacks in terms of accuracy.

3. please explain why there is no competition for node E when node Z is added in Figure 4 of Section 2.1. Why is no authentication, competition, or voting required? Why is there no authentication, competition or voting when node E joins?

4. In Figure 13, please explain how "Is the model updated?" is determined and executed? The flowchart is not perfect and not user friendly, so it is impossible to see what the author wants to express.

5. How can the author prove that the three data sets simulated in this article are in line with the real environment? Please explain.

6. In lines 287 to 292, the authors state that data balancing uses Random Over-Sampling (ROS) as the random sampling method, please provide relevant data to prove why data balancing can be achieved by this method.

Reviewer 2 Report

The paper needs proof-reading as the sentence ordering is wrong. Also, please fix / address the following issues: 

  • Missing dot (33)
  • to be crashed by packet (43)
  • Quality of service (QoS) (48)
  • in (Table 2)
  • as well as neither (199)
  • strictures (251)
  • where -> were (256)
  • is is (336)
  • are he (376)
  • As stated in Section , (415)
  • Eventough (425)
  • extensively use (443)
  • more tan (455)
  • wok (458)

Figures 9 a 10 could be easily combined together. 

Reviewer 3 Report

The merit of the proposed approach is supported by the results, but I miss on the paper a bit more discussion on why these techniques were chosen for this problem and had not been considered before. This however is more of a nitpicking than a detrimental comment.

What was the key motivation behind focusing on the Clone ID Attacks?

It would be interesting if the authors report the trade-off compared to other methods especially the computational complexity of the models. Some techniques require more memory space and take longer time, please elaborate on that.

Conclusion is too short. Add more explanation.

What are the limitations of the present work?

Authors should further clarify and elaborate novelty in their contribution.

What are the implications of this research?

Below papers has some interesting implications that you could discuss in your introduction and how it relates to your work.

 Ijaz, Muhammad Fazal, Muhammad Attique, and Youngdoo Son. "Data-Driven Cervical Cancer Prediction Model with Outlier Detection and Over-Sampling Methods." Sensors 20.10 (2020): 2809.

Ali, Farman, et al. "A smart healthcare monitoring system for heart disease prediction based on ensemble deep learning and feature fusion." Information Fusion 63 (2020): 208-222.

Round 2

Reviewer 1 Report

All the questions of the first version have been statements, but the authors should still detail the steps and process of resolving the competition between the legitimate node E and the malicious node Z.

Reviewer 3 Report

.
